# Rapid Prototyping of 3D-Printed AgNPs- and Nano-TiO_2_-Embedded Hydrogels as Novel Devices with Multiresponsive Antimicrobial Capability in Wound Healing

**DOI:** 10.3390/antibiotics12071104

**Published:** 2023-06-25

**Authors:** Giulia Remaggi, Laura Bergamonti, Claudia Graiff, Maria Cristina Ossiprandi, Lisa Elviri

**Affiliations:** 1Food and Drug Department, University of Parma, Parco Area delle Scienze 27/a, 43124 Parma, Italy; giulia.remaggi@unipr.it; 2Department of Chemistry, Life Sciences and Environmental Sustainability, University of Parma, Parco Area delle Scienze 17/A, 43124 Parma, Italy; 3Department of Veterinary Science, University of Parma, Strada del Taglio 10, 43126 Parma, Italy

**Keywords:** 3D printing, antimicrobial activity, hydrogels, silver nanoparticles, titanium dioxide

## Abstract

Two antimicrobial agents such as silver nanoparticles (AgNPs) and titanium dioxide (TiO_2_) have been formulated with natural polysaccharides (chitosan or alginate) to develop innovative inks for the rapid, customizable, and extremely accurate manufacturing of 3D-printed scaffolds useful as dressings in the treatment of infected skin wounds. Suitable chemical–physical properties for the applicability of these innovative devices were demonstrated through the evaluation of water content (88–93%), mechanical strength (Young’s modulus 0.23–0.6 MPa), elasticity, and morphology. The antimicrobial tests performed against *Staphylococcus aureus* and *Pseudomonas aeruginosa* demonstrated the antimicrobial activities against Gram+ and Gram− bacteria of AgNPs and TiO_2_ agents embedded in the chitosan (CH) or alginate (ALG) macroporous 3D hydrogels (AgNPs MIC starting from 5 µg/mL). The biocompatibility of chitosan was widely demonstrated using cell viability tests and was higher than that observed for alginate. Constructs containing AgNPs at 10 µg/mL concentration level did not significantly alter cell viability as well as the presence of titanium dioxide; cytotoxicity towards human fibroblasts was observed starting with an AgNPs concentration of 100 µg/mL. In conclusions, the 3D-printed dressings developed here are cheap, highly defined, easy to manufacture and further apply in personalized antimicrobial medicine applications.

## 1. Introduction

Skin is the largest organ in the human body and acute or chronic wounds, depending on duration of the healing, could lead to compromised health and immunity. Potentially, all wounds could become chronic (healing process lasting more than three weeks), especially in presence of pathologies such as diabetes, vascular insufficiency, and infections. Skin wound management represents a big challenge for the entire scientific and clinical community [1,2]. Economic and social impacts suggest that breakthrough technologies and approaches are necessary to meet the present and future needs of patients suffering from chronic skin wounds.

In this frame, functional biomaterials, regenerative medicine, and wound healing are deeply connected [3]. As antibiotic resistance is at dangerously high levels in all parts of the world, focusing on infected chronic wounds, regenerative medicine is moving toward cost-effective, efficient, and, in the ideal situation, patient-personalized solutions to treat microbial infection together with the stimulation of new tissue regeneration. The adverse impact on patients’ quality of life is paramount considering that, in the currently available treatments, no single approach can tackle all the challenges associated with wound healing, namely the high volume of exudates, microbial infection, and low perfusion. The most frequent pathogens present at the wound level are *Staphylococcus aureus* (Gram+; aerobic)—MRSA and *Pseudomonas aeruginosa* (Gram−; anaerobic); both species have different virulence factors that mediate adhesion, tissue destruction, avoidance of the immune system, and resistance to antibiotics. The latter aspect is of particular importance in hospital environments, where multiresistant strains are present leading to higher medical costs, prolonged hospital stays, and increased mortality [4]. These bacteria produce biofilms as physical barriers to the penetration of antimicrobial agents [5,6]; therefore, infection prevention is an important aspect to consider when designing dressings. The functionalization of biomaterials is commonplace nowadays, and the development of innovative antimicrobial medications able to not induce resistance and concurrently promoting tissue regeneration as advanced therapeutic solutions can be of great interest.

In this paper we aim at designing and manufacturing three dimensional (3D)-printed hydrogels obtained from biocompatible polymers, such as chitosan (CH) and alginate (ALG), and functionalized using silver and titanium dioxide nanoparticles (AgNPs and TiO_2_), for the healing of a large variety of locally infected wounds. Three-dimensionally structured dressings present bespoke features such as porosity (to allow the passage of oxygen), transparency (to allow the passage of light), and biodegradability (to allow the resorption of the device, if necessary). Moreover, different 3D-printed dressings can be easily manufactured in a personalized manner depending on the nature of the wound and the patient’s needs.

In general, polymeric hydrogels, thanks to their physical–chemical properties, are beyond the most used materials as scaffold: they prevent dehydration of the wound, thanks to the highly hydrophilic nature [7], they have low mechanical resistance but good elasticity, are the most similar biomaterials to soft tissues in consistency [8], they can absorb exudates [9], and they are biodegradable. Chitosan and alginate are natural polysaccharide able to form hydrogels, with some specific characteristics that make them particularly suitable for medical applications [10]. In particular, chitosan can present antimicrobial activity thanks to the interaction between the positive charges of chitosan and the surface of the negatively charged bacterial cells. These interactions promote the alteration of cell membrane permeability, leading to osmotic imbalances, loss of electrolytes, and low-molecular-weight cell components [11]. This polymer has been combined with different materials such as sulfadiazine [12], zinc oxide [13,14], titanium dioxide [14], and silver nanoparticles [15,16], with the aim of enhancing the antimicrobial activity [17].

Indeed alginate, compared to chitosan, has greater mechanical strength, a reduced ability to host cell cultures, and does not have antimicrobial activity [18]. ALG has already been used to make hydrogels, films, membranes, nanofibers and foams for the treatment of skin wounds [16], and associations with active substances have been investigated, such as zinc oxide [19,20], silver nanoparticles [5,20], and sulfadiazine [21], in order to confer antimicrobial activity to the dressing.

Silver nanoparticles release silver ions through an oxidation process, that occurs depending on various factors, such as the surface area, the concentration of oxygen in solution, and the dimensions: for the same quantity of silver, smaller nanoparticles have demonstrated increased activity [22]. Silver is a broad-spectrum antimicrobial agent, which uses various nonspecific mechanisms and is, therefore, suitable for overcoming the problem of antibiotic resistance. Toxicity is linked to the interaction of the Ag^+^ ion with the thiol groups of membrane proteins and enzymes involved in cellular respiration, and with DNA; the result is the inhibition of cell proliferation. Toxicity is also due to a second mechanism, dependent on the type of cell: the formation of oxygen free radicals [22].

Silver is also toxic to fibroblasts, in a dose-dependent way. It is, therefore, necessary to identify the therapeutic window that allows the prevention of infections and, at the same time, promotes healing [23].

Titanium dioxide, such as silver, uses nonspecific toxicity mechanisms. It is a semiconductor and photocatalytic material which is able, in the presence of UV radiation, to degrade organic compounds by catalyzing oxidation reactions. For this property, titanium dioxide is studied in order to develop self-cleaning materials, capable of eliminating polluting substances and microorganisms [24]. There are several crystalline forms of TiO_2_; the main ones are anatase, rutile, and brookite. Among these, anatase is the most effective photocatalyst [25]. Photocatalytic activity and the consequent production of ROS are at the basis of the toxicity mechanism: exposure to TiO_2_ causes an alteration of permeability in the cell membrane [26], a reaction of lipid peroxidation, and the inhibition of cellular respiration [27]. Different formulations containing TiO_2_ have been studied, to promote wound healing: titanium dioxide has been shown to promote coagulation [28] and to be compatible with various human cell lines, including fibroblasts [29].

To date, the creation of 3D-printed chitosan/alginate-based scaffolds containing silver and titanium dioxide nanoparticles has not been studied. For these reasons, in this paper, the research activity described was focused on a stepwise conceived strategy aimed to achieve the following: (1) To formulate of up to twelve CH or ALG innovative inks containing AgNPs and/or TiO_2_; (2) To produce and characterize 3D-printed hydrogels with suitable properties to be actively applied on skin wounds; (3) To test in vitro cell-viability performances on human fibroblasts; (4) To test in vitro antimicrobial performances on multidrug-resistant strains of *Staphylococcus aureus* and *Pseudomonas aeruginosa*.

## 2. Results

### 2.1. 3D-Printed Scaffold Preparation and Characterization

The creation of hydrogels with a well-defined 3D structure, starting from CH, ALG, AgNPs, and nanoTiO_2_, could be functional in order to obtain a variety of medications with antimicrobial properties.

In the first step, attention was focused on the development of the polysaccharide/nanoparticle-based inks allowing printability and the stable embedding of AgNPs and/or nano TiO_2_ in the highly structured 3D hydrogels without a time-consuming and poorly reliable postprocessing loading step. CH and ALG solutions (both at 6% *w*/*v*) were used as control inks, as already optimized in previous works, in terms of viscosity and continuous layer printability suitable to obtain hydrogels with final mechanical properties compatible for applications in wound healing [30,31]. The stability of AgNPs in the gelling agent solutions for CH and ALG (KOH 2% and CaCl_2_ 3% *w*/*v*, respectively) was demonstrated at different nanoparticle concentrations (0.05, 0.1, and 0.2 mg/mL) by performing spectrophotometric UV/VIS absorbance measurements at 400 nm over 60 min (*p* < 0.05) (data not shown). As an example, the trend of AgNPs absorbance in CaCl_2_ solution (3% *w*/*v*) is shown in Figure 1A.

The absorption spectrum of AgNPs, in the range between 300 and 500 nm (Figure 1B), shows how the position of the peak maximum in the ALG solution is shifted to 435 nm. The position of the peak, as described in the literature [32], is dependent on the size of the silver nanoparticles.

A peak shift at longer wavelengths could be attributable to an increase in size, which in this case could be due to an aggregation phenomenon in the polysaccharide solutions.

In the presence of hydroxides, silver forms a black precipitate of silver oxide (Ag_2_O). The scaffolds based on chitosan and silver nanoparticles, after one hour of permanence in KOH 2% *w*/*v*, did not exhibited any color variations. A gray color started to be visible only after several hours of permanence of the scaffolds in the gelling solution.

By considering that only a few minutes (up to 5 min for 20 layers scaffolds) are required for the complete cross-linking of the CH and ALG 3D-printed hydrogels, the stability of the AgNPs was considered suitable for the application.

Regarding the hydrogel manufacturing process, at macroporosity level, our home-made 3D-printing system [33,34] allowed control over the deposition trajectory of a filament with a diameter of 180 μm with an accuracy of ±10 μm for all the inks formulated and reported in Table 1. The complete control of the macropore size and number of printed layers was ensured, allowing the manufacture of patches with suitable thickness to fit every type of skin wound.

Chemical, physical, and mechanical characterizations via analytical and microscopic techniques including vibrational spectroscopies (FTIR), scanning electron microscopy (SEM), and mechanical resistance analysis were, thus, performed.

The determination of the water content is a very important parameter affecting both the biomechanical properties of hydrogels and compatibility with a living host tissue [5]. The highest water amount in the dressing gives indications about the wettability of the medication, the capability of absorbing exudates, and ability to keep the wound moist [5]. Figure 2 shows that the amount of silver nanoparticles does not alter the water content of both the CH (~92%) and ALG (~90%) scaffolds. On the other hand, both CH and ALG formulations containing the combination of nano silver and titanium dioxide added at the 1% (*w*/*v*) showed moderate reduction (~2%) in water absorptivity with respect to the polymer alone or with nano silver. Nano TiO_2_ is known to induce a reduction in porosity and, consequently, the surface/volume ratio of the hydrogel together with a reduction in its swelling degradation [35].

The mechanical properties of the materials have been characterized in order to assess their applicability on human tissues and, in particular, on skin. In the literature, the Young’s modulus of human skin varies between 4.6 and 20 Mpa in tensile tests [35]. Figure 3A compares the strength recorded at the moment of breakage, during the tensile test. Compared to chitosan, alginate-based hydrogels are characterized by greater mechanical strength, due to cross-links of an ionic nature. Chitosan, on the other hand, forms weaker bonds, in particular hydrogen bonds and hydrophobic interactions, and is, therefore, less resistant. The presence of silver nanoparticles, as well as that of titanium dioxide, did not significantly alter the mechanical strength of the ALG hydrogel.

Figure 3B compares the elasticity of the different materials, expressed as Young’s modulus (Mpa). The lower the Young’s modulus value, the greater the elasticity of the material. All formulations are characterized by greater elasticity than human skin, and can, therefore, be considered compatible. Furthermore, the presence of silver nanoparticles did not affect elasticity or resistance. On the other hand, the presence of titanium dioxide positively influenced the elasticity of chitosan. Alginate-based scaffolds, therefore, combine greater mechanical strength with less elasticity than chitosan-based scaffolds.

### 2.2. Scanning Electron Microscopy (SEM) and Energy-Dispersive X-ray Spectroscopy (EDS)

To compare the morphology of the different scaffold formulations and to study the distribution of silver and titanium nanoparticles in the scaffold matrix, SEM-EDS analyzes were performed.

Figure 4a–d shows the SEM images of the CH and ALG scaffolds (Figure 4a,b, respectively) and with the addition of titanium dioxide (Figure 4c,d, respectively).

Both scaffolds have a regular shape made up of filaments of about 150 μm intertwined to form a net, with fairly regular meshes of about 200 μm on each side. The surface of the scaffolds is characterized by a homogeneous roughness and diffused porosity, with pores of about 20 μm in diameter. The addition of TiO_2_ nanoparticles does not alter the scaffold structure, as evident in Figure 4c,d.

Figure 5a–f show SEM-EDS investigations performed on chitosan scaffolds with embedded AgNPs and TiO_2_. The secondary electron images (Figure 5a,d) show a well-defined morphology of the scaffolds, with filaments of 100 microns in size, crossed to form a grid.

The AgNPs are uniformly distributed and follow the scaffold structure, as evident from the elementary map of Ag reported in Figure 5b. In the scaffold made with formulation 4 (CH 6% *w*/*v*+ AgNPs 100 µg/mL + TiO_2_ 1% *w*/*v*) with AgNP and TiO_2_ (Figure 5d), the TiO_2_ is regularly distributed in the scaffold matrix, with rare agglomerates of nanoparticles, as clearly visible from the elemental map analysis (Figure 5e).

Chitosan has the ability to stabilize the shape and distribution of Ag NPs [36]. As for CH, the ALG-based formulations exhibited homogeneous distribution of nano silver and nano titanium dioxide in the final 3D-printed hydrogel. These findings allow us to conclude the homogeneity of the formulations used as 3D inks, which also remains in the final structure in the hydrogel state.

### 2.3. Antimicrobial Activity Assay

In order to evaluate the antimicrobial efficacy of the developed devices, scaffolds were compared in terms of the inhibition of bacterial growth in bacteria-inoculated Petri dishes after 24 h (Table 1). The antimicrobial activity test demonstrated the effectiveness of the chitosan control scaffolds against *Pseudomonas aeruginosa* and *Staphylococcus aureus*, confirming the antimicrobial properties of this polysaccharide. Different AgNP concentrations were tested to assess MIC, when embedded in 3D chitosan- or alginate-based scaffolds. Chitosan–AgNP-based scaffolds showed an increase in the inhibition diameter (>6 mm) in a dose-dependent manner, attributable to release and diffusion of silver in the culture medium and analogous for both bacterium strands (Table 1). The addition of titanium dioxide only at the 1% *w*/*v* in the chitosan scaffold did not significantly alter the antimicrobial efficacy of the chitosan alone. This result can be explained by considering two different factors: the first one could be a possible low diffusion of TiO_2_ through the chitosan hydrogel; the second one is that the experiments were carried out without any photoactivation of the TiO_2_. The simultaneous presence in the CH scaffold of both AgNPs (100 μg/mL) and TiO_2_ (1% *w*/*v*) did not modified the antimicrobial activity against *S. aureus* but significantly reduced the activity against *P. aeruginosa*.

The test performed on alginate-based scaffolds, which do not possess antimicrobial activity alone, have allowed us to identify the minimum inhibitory concentration relating to silver nanoparticles, corresponding to 5 μg/mL for *P. aeruginosa* and 10 μg/mL for *S. aureus*. The antimicrobial activity of titanium dioxide alone has been confirmed in the ALG hydrogel and, under the experimental conditions adopted, did not show any synergistic effects with AgNPs.

### 2.4. In-Vitro Cytocompatibility Tests

Cytocompatibility is of relevance for specific cell growth in the healing process. As a last application, human fibroblasts were grown on scaffolds in order to evaluate their biocompatibility.

Initially, the biocompatibility of the chitosan and alginate scaffolds free of antimicrobial agents were tested over 28 days (Figure 6). Compared to chitosan, which is considered biocompatible compared to control (Petri dish without any scaffold) [37], alginate demonstrates less compatibility with the growth of fibroblasts, confirming data already reported in the literature [37,38]. After 14 days alginate scaffolds exhibited a 50% reduction in cell viability, which remained constant over 28 days. Figure 7A,B compares the cell viability of the CH and ALG hydrogels containing AgNPs and/or TiO_2_ materials, respectively. The cell viability data of the different formulations containing antimicrobial agents were normalized with respect to cell viability measured on the chitosan polymer-alone scaffolds, due to its proven compatibility with human fibroblasts. The data relating to chitosan (Figure 7A) demonstrated that the silver nanoparticles, at the concentration of 0.01 mg/mL did not significantly alter the viability of fibroblasts over 28 days. When the AgNPs concentration was at 0.1 mg/mL, cell viability was negatively affected over time with a 50% reduction over 21 days, as well as in the presence of TiO_2_. The ALG formulations containing silver nanoparticles and TiO_2_ exhibited similar trends as a function of time, with a significantly decrease in cell viability with respect to the reference alginate alone after 7 days. The presence of TiO_2_ has a marginal influence on cell viability: this effect could be explained by considering that, in the absence of exposure to light, titanium dioxide is free of catalytic activity.

These data confirmed the greater biocompatibility of chitosan compared to alginate and some toxicity effects of AgNPs. In particular, the critical factor is the concentration of the silver nanoparticles which, at the concentration of 0.01 mg/mL, did not cause a significant reduction in cell viability, while maintaining antimicrobial activity, as demonstrated by tests on *S. aureus* and *P. aeruginosa*. However, it should be highlighted that the 3D-printed hydrogel dressing manufactured here allowed us to obtain a not significant toxicity of silver nanoparticles towards human cells within a time window of 7 days. By considering that in wound care, dressings need to be changed every one to three/four days; these hydrogels can be more than suitable for a potential use in in vivo antimicrobial tests.

## 3. Discussion

Natural polymers (chitosan and alginate) and antimicrobials agents (AgNPs and TiO_2_) inks have been formulated to manufacture novel highly defined 3D crio-printed scaffolds useful as dressings in wound care. The characterization in terms of water content, mechanical strength, elasticity, and morphology of the developed devices demonstrated their suitable CHEMICAL-physical properties for application. In particular, they have a high-water content (>86%), important for the maintenance of a hydrated environment at the wound level, a condition that promotes the healing process, and an elasticity superior to that of the skin. The results of antimicrobial tests against *Staphylococcus aureus* and *Pseudomonas aeruginosa* demonstrated that the antimicrobial agents in the formulations are effective at least for 24 h in preventing infections caused by these bacteria often present within infected wound. The 3D-printed structure allowed for an high surface-to-volume ratio that can improve the diffusivity of active compounds, bacteria–biomaterial adhesion, and their wall disruption. This can be an excellent advantage for the treatment of deep wound with hydrogels of variable thickness.

Cell viability test showed that the materials based on chitosan are characterized by a greater biocompatibility, compared to those based on alginate. Silver nanoparticles have been shown to be cytotoxic towards human fibroblasts in a dose and time-dependent manner, whereas the presence of titanium dioxide did not significantly modify the activities of the scaffolds against cells. Chitosan proved to be the most suitable material for making dressings, being endowed with intrinsic antimicrobial activity and greater biocompatibility with respect to alginate. However, alginate was better with regard the mechanical properties and, since in wound care dressings need to be changed within three/four days, both biomaterials can be more than suitable for antimicrobial use. As key point, we can state that it is reasonable to think that the antimicrobial properties of TiO_2_-embedded hydrogels could be further enhanced by exploiting the photoactivation potential of this support, that was already demonstrated for environmental applications [39]. Some shortcomings can be present, such as poor absorption of wound exudate by wet dressings; the antibacterial activity against *S. aureus* and *P. aeruginosa* should be tested even with the dried hydrogels and for longer than 24 h.

The near future will be characterized by an increase of the development of innovative dressings able to provide advantages in terms of efficacy and costs in a patient-personalization therapeutic frame. The advances in technology, including 3D printing is of utmost importance and will be the cutting edge of modern dressings. The number of natural polysaccharide-based dressing devices is growing as they have been demonstrated to be useful to produce reliable results. Once verified by in vivo experiments and clinical trials, it is reasonable to think that the demand for such products could increase, leading to the commercialization of such products.

## 4. Conclusions

With this discussion we can conclude that the 3D-printed dressings developed here own efficient and tunable antimicrobial features (i.e., different thickness of the hydrogel, different loaded amount of antimicrobial agents, and controlled delivery of active compounds). Three-dimensional printing is moving hand in hand with medicine requests toward the easy, cheap, and reliable manufacturing of personalized and more sophisticated dressing useful within routine life.

## 5. Materials and Methods

### 5.1. Materials

The sodium alginate (Ph.Eur. grade; molecular weight by gel filtration chromatography (GFC) 180–300 kDa; slowly soluble in water), was from Carlo Erba (Carlo Erba Reagents Srl, Milan, Italy). Chitosan ChitoClear TM4830, with a degree of deacetylation of 75% and a molecular weight of 50 kDa, was obtained from Primex (Primex EHF, Siglufjordur, Iceland).

AgNP suspension (nominal concentration: 1000 ppm; polydispersity index 0.4) was prepared by the reduction of AgNO_3_ in water with NaBH_4_, following a procedure previously described [39]. The acetic acid 99.8% 10L150515 was from VWR (VWR International GmbH, Darmstadt, Germany); the titanium dioxide Aeroxide^®^ P25, nanopowder, 21 nm primary particle size (MKBV3126V) was from Aldrich (Merck, St. Louis, MI, USA); the calcium chloride anhydrous 419887/1 was from Fluka Chemie GmbH (Fluka Chemie GmbH, Buchs, Switzerland); ethanol (96% *v*/*v*); the potassium hydrate P0119208 was from ACEF (ACEF Spa, Piacenza, Italy); the sodium tribasic citrate dihydrate 1986C100 Codex was from Carlo Erba, (Italy); and the EDTA 61930 was from Riedel-de Haen (Riedel-de Haën GmbH, Seelze, Germany).

### 5.2. Methods

#### Ink Preparation for 3D Printing

The preparation of ink formulations suitable for 3D printing was carried out by taking into account of the crucial solution parameters such as viscosity, polymer concentrations, and homogeneity [40,41,42]. In particular, up to twelve different inks were prepared as described in Table 2.

A suspension of AgNPs was prepared at the desired concentration. The chitosan powder alone or with titanium dioxide powder were initially dispersed in the suspension, by stirring on a magnetic plate. Subsequently, acetic acid was added drop by drop to dissolve the chitosan. The formulation was kept away from light and stirring until a homogeneous mixture was obtained.

The alginate powder was added to the suspension of AgNPs alone or with TiO_2_ while stirring on a magnetic plate. Once ready to use, the formulations were stored at 4 °C in the dark light to avoid possible interactions between light and silver nanoparticles and to prevent the activation of titanium dioxide. Spectrophotometric UV/Vis analyses were carried out using a Cary instrument from Agilent (Agilent Technology, Santa Clara, CA, USA).

### 5.3. 3D Printing and Scaffold Production

An home-made 3D printer was specifically built to print aqueous viscous materials for the accurate production of hydrogel scaffolds [30,39]. Briefly, the ink (viscosity range 8–40 kcP) is printed layer by layer on a stainless-steel surface plate cooled at −14 °C, that instantaneously solidifies the extruded filaments. The plate on which the scaffold is deposited is removed at the end of the printing process and immersed in the gelling solution (KOH 2% *w*/*v* for chitosan based formulations and CaCl_2_ 3% *w*/*v* for the alginate based formulations) where it remains for 5 min. At the end of the gelling process, which irreversibly confers the three-dimensional structure, the scaffolds were washed with ultrapure water for the removal of cross-linking excesses and stored at 4 °C.

### 5.4. Scaffold Characterization

Gravimetric analysis was used to evaluate the hydrogel water content. Briefly, scaffolds (20 layers; 1.4 cm × 1.4 cm; *n* = 5) of each type of ink were first tamponed on filter paper to remove the excess of water and then weighted to determine the wet weight (W_w_). After essication in oven at 40 °C the dry weight (D_w_) was measured and the % of water content was determined as follows (Equation (1)):100 − (100 × D_w_)/W_w_(1)

The elasticity (Young’s modulus) and elongation % at break were then measured to evaluate the mechanical properties of each hydrogel. The tests were performed on scaffolds (20 layers; 4 cm × 1.4 cm; *n* = 3) as previously reported [30,39] by using a tractional dynamometer (AG M1, Aquati Srl, Arese, Milan, Italy) (distance between clips: ± 25 mm, traction speed 25 mm/min, 5 DaN top head). A PowerLab 400 board (ADInstruments Ltd., Oxford, UK) and Scope 3.5 software (NI-Scope, Austin, TX, USA) were used to record the force applied by the tensile tester (N) and net movement (mm). Elongation at break (% strain) and Young’s modulus were also calculated (*n* = 3) [30]. In detail, Young’s modulus was calculated using the formula (2):E = σ/ε(2)
where σ is the applied force/cross section area (stress) and ε the net elastic elongation(strain). The ratio (100ε)/specimen length indicated the elongation percentage.

### 5.5. SEM and SEM-EDS Analysis

Silver nanoparticles and titanium dioxide inside the scaffolds were characterized within 5-layer scaffolds (1.4 cm × 1.4 cm) by following a preparation procedure already described [39]. Briefly, hydrogels were dehydrated until absolute ethanol that was then eliminated by employing Critical Point Drying (Balserz Union, Lake Butler, FL, USA) (70 atm, 37 °C). This procedure allows for the retention of the 3D structures that were accurately cut to obtain pieces exposing both the surface area and cross sections. They were then fixed on support using A double-sided carbon tape was used to immobilize the anhydrous scaffolds before sputtering coated (E5100, Polaron, Quorum Technologies Ltd., Leves, UK) with gold (thickness 60 nm) to optimize electrical conduction before evaluation through a Philips 501 SEM (Philips, Eindhoven, The Netherlands) at magnifications ranging from 150× to 320×. The distributions of silver nanoparticles and titanium dioxide in carbon coated scaffolds made using formulations 3 and 4 (Table 2) were determined using a scanning electron microscope Jeol JSM 6400 (Jeol Spa, Milan, Italy) equipped with an Oxford Instruments Link Analytical Si (Li) energy-dispersive system detector (SEM-EDS). Digital photographs of the hydrogels were analyzed by means of ImageJ software v. 1.53 (National Institute of Health, NIH, Bethesda, MD, USA).

### 5.6. Antimicrobial Activity Tests

Antimicrobial activity against multidrug-resistant strains of *Staphylococcus aureus* (ATCC 25923) (Manassas, VA, USA) and *Pseudomonas aeruginosa* (ATCC 27853) (Manassas, VA, USA) was assessed. Two strains of bacteria (Gram+ and Gram−) frequently responsible for infections in chronic wounds were considered [43].

The antimicrobial activities of the 3D-printed hydrogel scaffolds were assayed by adopting the diffusion disk method (or Kirby-Bauer technique) [44]. Five-layer scaffolds were created to evaluate the antimicrobial activity of the chitosan-based formulations and were punched to obtain 6 mm diameter discs. For alginate-based formulations, which are subject to degradation inside the culture medium, disks of the same size were obtained from 15-layer scaffolds. After that, samples were sterilized in 70% *v/v* ethanol [40], rinsed, and stored in sterile water at 4 °C until use.

The formulations reported on Table 2 have been tested in two replicates for each bacterial strain.

A Mueller Hinton Broth at 37 °C was used to inoculate bacteria under aerobic conditions (1–2 h; 0.5 McFarland). Bacterial suspension in pure culture was seeded through the use of sterile tampons on a Mueller Hinton Agar terrain (carefully covering the entire Petri dish). After deposition of sterile scaffolds, all the plates were incubated at 37 °C for 24 h. The bacterial sensibility to antimicrobial agents was proportionally correlated to the absence/presence and diameter determination of the inhibition ring [45].

### 5.7. Cell Viability Test

Ten-layer scaffolds were printed on films of the same material, to prevent contact between the cells and the bottom of the wells. The 6 mm diameter disks were made from the scaffolds, which were sterilized in 70% *v/v* ethanol and rinsed in sterile PBS 1X. The scaffolds were arranged in 48-well plates.

After aspirating the medium and washing with PBS, the cells were detached from the plates with a trypsin/EDTA solution (0.25% *w*/*v* trypsin/1 mM EDTA) and seeded at the density of 235,000 cells per scaffold. Human fibroblasts have been used. After one hour, the culture medium (DMEM + 10% FBS) was added, which was replaced every 3 days.

1 mg/mL sodium resazurin solution was prepared in PBS, as indicator for the evaluation of cell viability, and filtered on a 0.22 micron filter to sterilize and eliminate any undissolved precipitates. The resazurine was diluted 1:100 in HBSS (or medium without serum and without phenol red) to have a final concentration of 10 mg/mL; the medium was aspirated from the cells that were washed with PBS. Resazurin solution (300 μL for 48 well plates) was added and the samples were incubated for 2 h at 37 °C. Finally, the fluorescence with the fluorimeter set with excitation at 540 nm and emission filter at 590 nm was evaluated. By means of the resazurin test, the metabolic activity of fibroblasts sown on scaffolds containing silver nanoparticles was compared with that of cells sown on control scaffolds, free of nanoparticles. The following scaffolds reported on Table 3 have been used to assess cellular activity after 7, 14, 21, and 28 days. Three independent measurements were performed for each scaffold.

### 5.8. Statistical Analysis

Statistical analysis was carried out using Microsoft Excel software v. 16.68 (Microsoft Corporation, Redmond, WA, USA). Data are given as mean ± standard deviation (SD). A value of *p* < 0.05 was considered statistically significant.

## Figures and Tables

**Figure 1 antibiotics-12-01104-f001:**
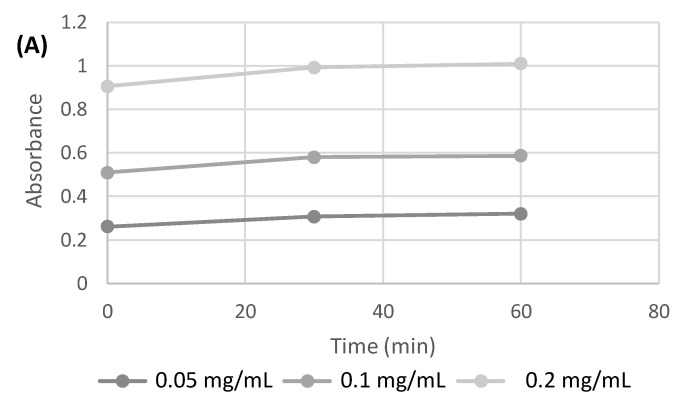
(**A**) Absorbance measured at 400 nm over time for AgNPs at different concentrations in CaCl_2_ (3% *w*/*v*) solution. (**B**) Absorption spectrum of AgNPs in aqueous standard solution and in ALG solution.

**Figure 2 antibiotics-12-01104-f002:**
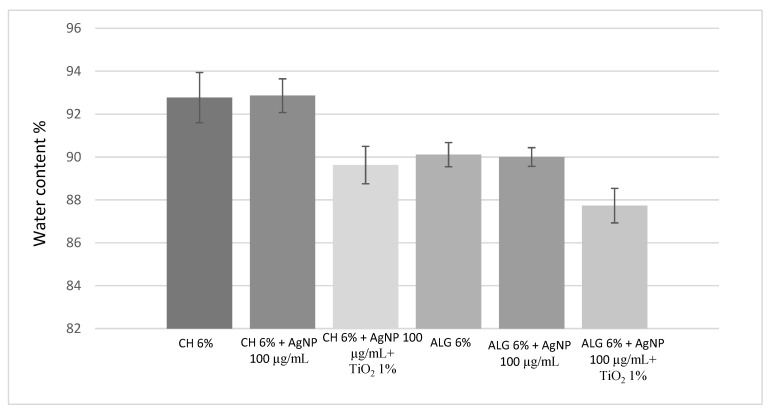
Water content of the 3D-printed scaffolds obtained using the six different inks. The error is expressed as relative standard deviation (*n* = 3).

**Figure 3 antibiotics-12-01104-f003:**
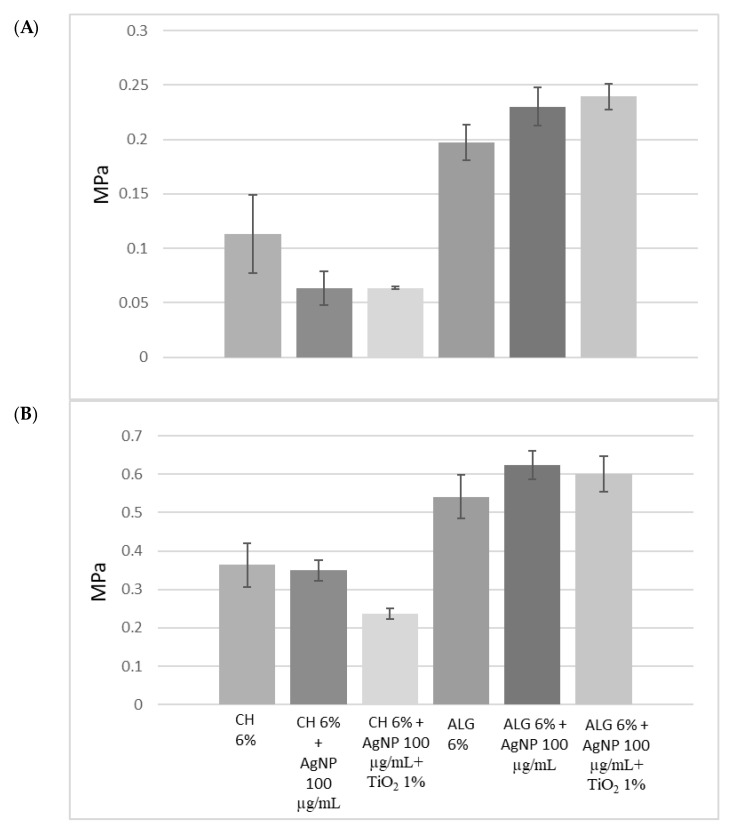
(**A**) Mechanical resistance, expressed as a stress applied to the breaking point; (**B**) Young’s modulus of the 3D-printed scaffolds obtained using the six different inks. The data are obtained from the average of three replicates and the error is expressed as a standard deviation.

**Figure 4 antibiotics-12-01104-f004:**
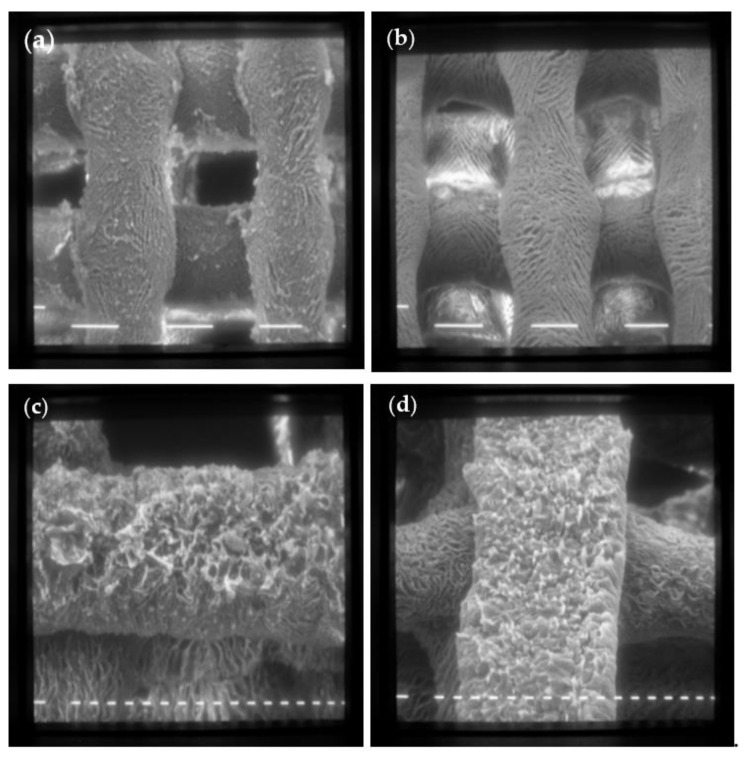
SEM images of scaffolds made with formulation 1 (CH 6% *w*/*v*) (**a**) and formulation 6 (ALG 6% *w*/*v*) (**b**) at 80× magnification (scale bar 100 μm); (**c**) Formulation 5 (CH 6% *w*/*v* + TiO_2_ 1% *w*/*v*); (**d**) Formulation 12 (ALG 6% *w*/*v* + TiO_2_ 1% *w*/*v*) at 160× magnification (scale bar 20 μm).

**Figure 5 antibiotics-12-01104-f005:**
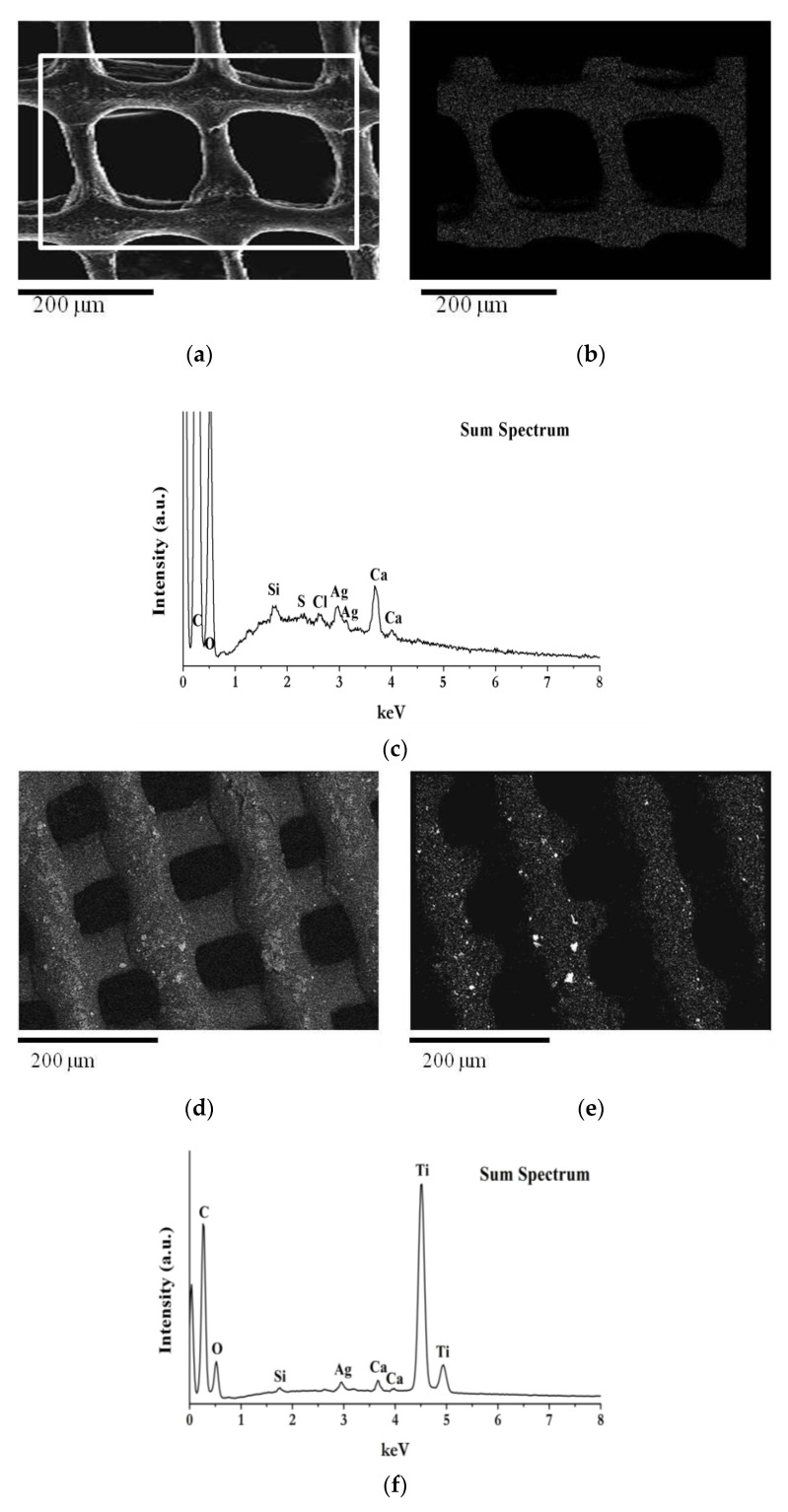
SEM image at 150× magnification and EDS maps: (**a**) Secondary electron image of scaffold made with formulation 3 (CH 6% *w*/*v* + AgNPs 100 µg/mL); (**b**,**c**) EDS silver map and EDS elemental analysis acquired in the selected area of (**a**); (**d**) Backscattered electron image of scaffold made with formulation 4 (CH 6% *w*/*v* + AgNPs 100 µg/mL + TiO_2_ 1% *w*/*v*); (**e**,**f**) EDS Ti map and EDS elemental analysis acquired over the whole area of (**d**).

**Figure 6 antibiotics-12-01104-f006:**
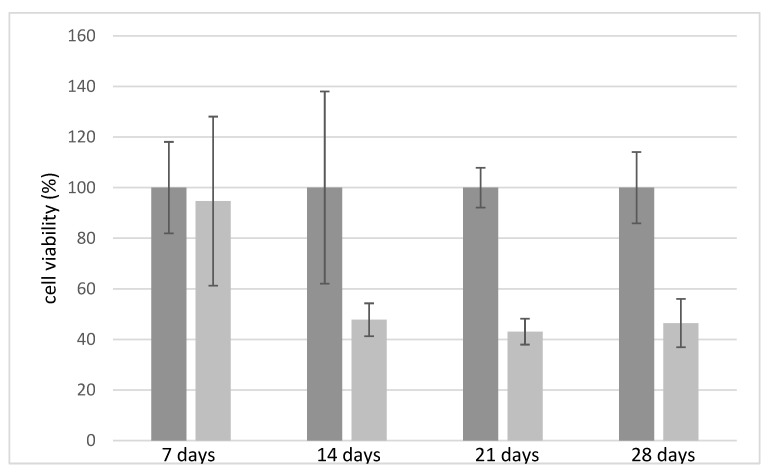
Relative fibroblast cell viability on chitosan (6% *w*/*v*) (in gray) and alginate (6% *w*/*v*)-based scaffolds (in light grey) tested over time.

**Figure 7 antibiotics-12-01104-f007:**
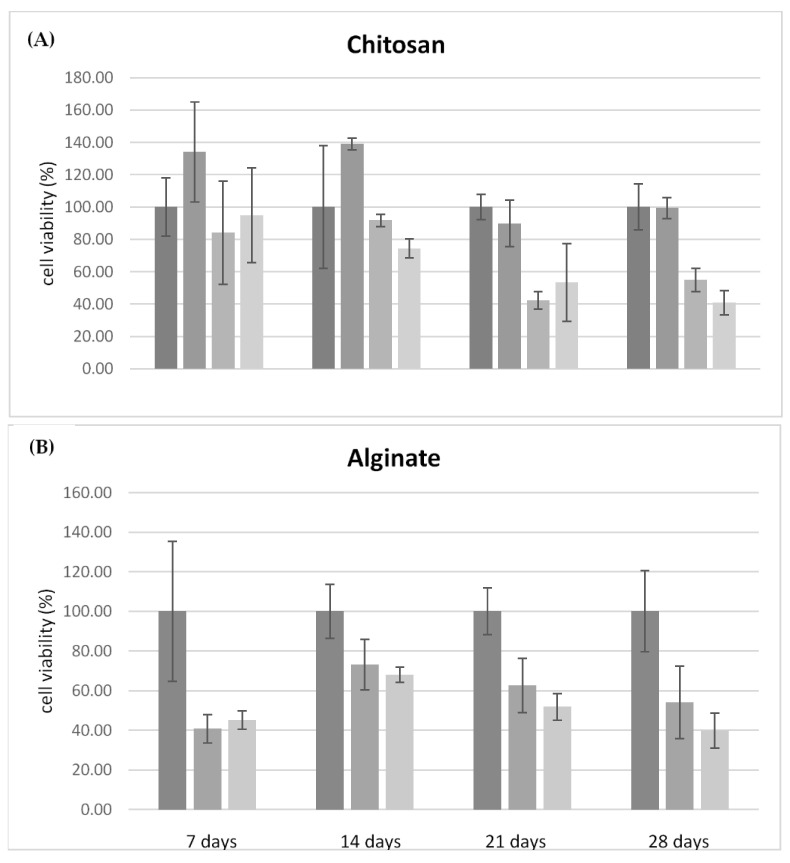
(**A**) Relative cell viability on chitosan-based materials, compared to the formulation of chitosan alone, tested over 28 days. Formulations tested: CH (6% *w*/*v*); CH (6% *w*/*v*) + AgNP (10 μg/mL); CH (6% *w*/*v*) + AgNP (100 μg/mL); and CH (6% *w*/*v*) + AgNP (100 μg/mL) + TiO_2_ (1% *w*/*v*). (**B**) Relative cell viability on alginate materials, compared to the formulation of alginate alone tested over 28 days. Formulation tested: ALG (6% *w*/*v*); ALG (6% *w*/*v*) + AgNP 100 μg/mL; and ALG (6% *w*/*v*) + AgNP (100 μg/mL) + TiO_2_ (1% *w*/*v*).

**Table 1 antibiotics-12-01104-t001:** Determination of antimicrobial activity of 3D-printed developed scaffolds.

SCAFFOLD (Ø 6 mm)	*Staphylococcus aureus*		*Pseudomonas aeruginosa*	
	Ø Inhibition Diameter (mm)			
CH 6% *w*/*v*	6	6	6	6
CH 6% *w*/*v* + AgNP 10 μg/mL	7	7	7	7
CH 6% *w*/*v* + AgNP 100 μg/mL	8	8	8	8
CH 6% *w*/*v* + AgNP 100 μg/mL + TiO_2_ 1% *w*/*v*	8	8	6	6
CH 6% *w*/*v* + TiO_2_ 1% *w*/*v*	6	6	6	6
ALG 6% *w*/*v*	0	0	0	0
ALG 6% *w*/*v* + AgNP 100 μg/mL	6	6	8	8
ALG 6% *w*/*v* + AgNP 10 μg/mL	6	6	8	8
ALG 6% *w*/*v* + AgNP 5 μg/mL	0	0	6	6
ALG 6% *w*/*v* + AgNP 1 μg/mL	0	0	0	0
ALG 6% *w*/*v* + AgNP 100 μg/mL + TiO_2_ 1% *w*/*v*	6	6	6	6
ALG 6% *w*/*v* + TiO_2_ 1% *w*/*v*	6	6	6	6

**Table 2 antibiotics-12-01104-t002:** Composition of the inks formulated and used for the manufacturing of the corresponding 3D-printed scaffolds.

Ink	Polysaccharide (*w*/*v*)	AgNPs (μg/mL)	TiO_2_ (*w*/*v*)
1	Chitosan 6% (ctrl)	-	-
2	Chitosan 6%	10	-
3	Chitosan 6%	100	-
4	Chitosan 6%	100	1%
5	Chitosan 6%	-	1%
6	Alginate 6% (ctrl)	-	-
7	Alginate 6%	1	-
8	Alginate 6%	5	-
9	Alginate 6%	10	-
10	Alginate 6%	100	-
11	Alginate 6%	100	1%
12	Alginate 6%	-	1%

**Table 3 antibiotics-12-01104-t003:** Composition of the 3D-printed scaffolds used to test cell viability.

Ink	Polysaccharide (*w*/*v*)	AgNPs (μg/mL)	TiO_2_ (*w*/*v*)
1	Chitosan 6% (ctrl)	-	-
2	Chitosan 6%	10	-
3	Chitosan 6%	100	-
4	Chitosan 6%	100	1%
5	Chitosan 6%	-	1%
6	Alginate 6% (ctrl)	-	-
10	Alginate 6%	100	-
11	Alginate 6%	100	1%

## Data Availability

The authors declare that the data generated and analyzed during this study are included in this published article. In addition, datasets generated and/or analyzed during the current study are available from the corresponding author on reasonable request.

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
