# Peer review of "Rapid Prototyping of 3D-Printed AgNPs- and Nano-TiO2-Embedded Hydrogels as Novel Devices with Multiresponsive Antimicrobial Capability in Wound Healing"

_antibiotics, 2023, doi:10.3390/antibiotics12071104_

Round 1
Reviewer 1 Report
3D printed AgNPs and nano-TiO2 embedded hydrogels (chitosan and alginate) were developed as scaffolds to be used as dressings in the treatment of infected skin wounds. A series of tests have been performed to characterize the samples, including water content, particle size (UV-VIS Spectrometer), mechanical tests, surface morphology and element (SEM and SEM-EDS), as well as antimicrobial activity and cell viability tests. The overall experimental design and analysis are reasonable, the manuscript is also well referenced.
There are some minor problems:
1. The figure numbers are wrong in line 232, 234 and 235
2.
Language needs to be polished
Grammar problems: at least: line 32, 70, 71, 90, 108, 136, 236, 238, 286, 458
Author Response
We thank the Referee for the comments he provided and the good evaluation of our paper.
In agreement with the comments we corrected the figure numbers in line 232, 234 and 235
We improved the language and we corrected the grammar as indicated (line 32, 70, 71, 90, 108, 136, 236, 238, 286, 458).
Reviewer 2 Report
The manuscript addresses a highly interesting and significant topic, and it is commendably well-written and organized. However, there are a few areas that could benefit from improvement based on the referee's suggestions:
- The referee advises checking the bacteria names, specifically "Staphylococcus Aureus" and "Pseudomonas Aeruginosa".
- The Introduction paragraph is quite lengthy, and it would be beneficial to rephrase it while retaining the main points. Consider condensing it to provide a concise and focused introduction to the topic.
- The aim of the study needs to be clarified and emphasized better (lines 114-120). Please express in italics "in vitro".
- The discussion section would benefit from more attention. Consider incorporating future outcomes or discussing the limitations of the study to provide a more comprehensive analysis.
Lines 332-335 appear to resemble a final discussion rather than a conclusion. It may be more appropriate to create a subsection labeled "Conclusions" after that section.
- Taking these suggestions into account will strengthen the manuscript further.
Author Response
We really appreciate the Reviewer comments that help us to improve the manuscript.
Please find here our point-to-point answers.
- The referee advises checking the bacteria names, specifically "Staphylococcus Aureus" and "Pseudomonas Aeruginosa".
- We checked all over the text for the bacteria names.
- The Introduction paragraph is quite lengthy, and it would be beneficial to rephrase it while retaining the main points. Consider condensing it to provide a concise and focused introduction to the topic.
- The introduction was reduced to better summarize the information.
- The aim of the study needs to be clarified and emphasized better (lines 114-120). Please express in italics "in vitro".
- We improved the sentences highlighting the aim of the work. The term "in vitro" was expressed in italics.
- The discussion section would benefit from more attention. Consider incorporating future outcomes or discussing the limitations of the study to provide a more comprehensive analysis.
- The discussion section was improved in agreement with this comment.
Lines 332-335 appear to resemble a final discussion rather than a conclusion. It may be more appropriate to create a subsection labeled "Conclusions" after that section.
We tried to propose clearly both the discussion part and the conclusions.
Reviewer 3 Report
Comments to Authors:
The research article entitled “Rapid prototyping of 3D printed AgNPs and nano-TiO2 embedded hydrogels as novel devices with multi-responsive antimicrobial capability in wound healing” is very interesting approach. The manuscript is written well and its presentation is also good. This research work merit publishing however some major revisions and changes are needed.
1. Abstract: Revise the keywords list and arrange them alphabetically.
2. Abstract: Start the abstract first line with the background of the study
3. Abbreviations: The abbreviations should be clarified in first instance in abstract and rest of manuscript.
4. Improve the resolution of Figure 1, 2 and 3,4,5(f), 6 and 7 respectively?
5. Introduction: Add some latest literature citations i.e 2022,2023
6. Methods: All the equations including % decolorization should be assigned numbers as: (1) (2) and soon.
7. Methods: Numbering to sub section is necessary like 2.1 ,2.2,2.3 etc
8. Results: Numbering to sub section is necessary like 3.1 ,3.2,3.3 etc
9. Discussion: This section seems too short. Discuss your results with previous latest literature?.
10. Figur 7(A) Why the standard deviation in term of error bar is too high in 7 days. Also in Figure 6 standard deviation values are too high in 7 days and 14 days ?
11. References: The references should be revised according to journal format.
12. Manuscript should be revised carefully .There are some minor grammatical mistakes.
13. Conclusions: This section is missing why? This section should be included
14. How this research is practically valuable and beneficial?
15. The authors should check plagiarism of manuscript after revision.
16. Please follow the Instruction for authors.
Minor editing of English language required
Author Response
We thank the Reviewer for the comments and we adredded them to improve our manuscript.
- Abstract: Revise the keywords list and arrange them alphabetically.
Kewwords were modified according to this comment.
- Abstract: Start the abstract first line with the background of the study.
The Abstract was written by providing the aim of the work and the relevant data obtained.
- Abbreviations: The abbreviations should be clarified in first instance in abstract and rest of manuscript.
Abbreviations were clarified in the abstract first.
- Improve the resolution of Figure 1, 2 and 3,4,5(f), 6 and 7 respectively?
The figures are at the best resolution we are able to obtain. We hope they can be suitable for publication.
- Introduction: Add some latest literature citations i.e 2022,2023
Recent citations were added in the introduction section.
- Methods: All the equations including % decolorization should be assigned numbers as: (1) (2) and soon.
- Methods: Numbering to sub section is necessary like 2.1 ,2.2,2.3 etc
- Results: Numbering to sub section is necessary like 3.1 ,3.2,3.3 etc
Numbers were added to clarify the text and its organization.
- Discussion: This section seems too short. Discuss your results with previous latest literature?.
The manuscript was better organized and the discussion was improved.
- Figur 7(A) Why the standard deviation in term of error bar is too high in 7 days. Also in Figure 6 standard deviation values are too high in 7 days and 14 days ?
The error bars consider all the variability connected to the experiments that can reasonably be associated to overall handling precision.
- References: The references should be revised according to journal format.
The references were revised according to the journal format.
- Manuscript should be revised carefully .There are some minor grammatical mistakes.
The text was checked al over the manuscript.
- Conclusions: This section is missing why? This section should be included
Brief Conclusions were added.
- How this research is practically valuable and beneficial?
The potentil benefits of this research was better described in the final discussion.
- The authors should check plagiarism of manuscript after revision.
Plagiarism was checked over the text.
- Please follow the Instruction for authors.
The instructions for Authors were followed.
Reviewer 4 Report
The article discusses the development of innovative inks for the rapid, customizable, and extremely accurate manufacturing of 3D printed scaffolds useful as dressings in the treatment of infected skin wounds. The article also highlights the antimicrobial activities against Gram+ and Gram– bacteria of silver nanoparticles (AgNPs) and titanium dioxide (TiO2) agents embedded in the CH or ALG macroporous 3D hydrogels. The 3D printed dressings developed own the features to be cheap, highly defined, easy to be manufactured and further applied in personalized antimicrobial medicine applications.
The manuscript is well-written and well-organized, and the data supports the conclusions. However, there are some areas that require improvement in the manuscript.
The authors are advised to improve the quality of their figures by ensuring consistent font format, font size, and color throughout. This will enhance the appearance and readability of the figures.
When abbreviations such as "CH" and "ALG" are first mentioned in the abstract, it is recommended to provide their full names.
The term "stair bar" should be corrected to "scale bar".
It is unclear whether the dash line represents the scale bar as it does not seem to match the magnification mentioned. In Figure 4.
Figure 5c and 5f would benefit from being enlarged to improve clarity.
Considering the extended duration of 28 days for cell viability testing, it is expected that a significant number of cells would die due to overgrowth.
Minor editing of English language required
Author Response
The manuscript is well-written and well-organized, and the data supports the conclusions. However, there are some areas that require improvement in the manuscript.
We are grateful to the reviewer for these comments and we appreciate the opportunity to improve our manuscript.
The authors are advised to improve the quality of their figures by ensuring consistent font format, font size, and color throughout. This will enhance the appearance and readability of the figures.
In agreement with the reviewer's comment, we tried to improve at our best the quality of the figures.
When abbreviations such as "CH" and "ALG" are first mentioned in the abstract, it is recommended to provide their full names.
The full names of each acronym were provided.
The term "stair bar" should be corrected to "scale bar".
The term was corrected.
It is unclear whether the dash line represents the scale bar as it does not seem to match the magnification mentioned. In Figure 4.
The scale bare of figure 4 was checked and properly corrected.
Figure 5c and 5f would benefit from being enlarged to improve clarity.
The quality of figure 5c and 5f was improved.
Considering the extended duration of 28 days for cell viability testing, it is expected that a significant number of cells would die due to overgrowth.
Weunderstand this comment of the referee abut looking at the controls no cell death was observed over 28 days, due to a possible overgrowth.